# A fast and effective detection framework for whole-slide histopathology image analysis

**Jun Ruan[1], Zhikui Zhu[1], Chenchen Wu[1], Guanglu Ye[1], Jingfan Zhou[1], Junqiu Yue[2]***

**1** School of Information Engineering, Wuhan University of Technology, Wuhan, China, **2** Department of Pathology, Huazhong University of Science and Technology, Tongji Medical College, Hubei Cancer Hospital, Wuhan, China

* yuejunqiu@hotmail.com

## Abstract

Pathologists generally pan, focus, zoom and scan tissue biopsies either under microscopes or on digital images for diagnosis. With the rapid development of whole-slide digital scanners for histopathology, computer-assisted digital pathology image analysis has attracted increasing clinical attention. Thus, the working style of pathologists is also beginning to change. Computer-assisted image analysis systems have been developed to help pathologists perform basic examinations. This paper presents a novel lightweight detection framework for automatic tumor detection in whole-slide histopathology images. We develop the Double Magnification Combination (DMC) classifier, which is a modified DenseNet-40 to make patch-level predictions with only 0.3 million parameters. To improve the detection performance of multiple instances, we propose an improved adaptive sampling method with superpixel segmentation and introduce a new heuristic factor, local sampling density, as the convergence condition of iterations. In postprocessing, we use a CNN model with 4 convolutional layers to regulate the patch-level predictions based on the predictions of adjacent sampling points and use linear interpolation to generate a tumor probability heatmap. The entire framework was trained and validated using the dataset from the Camelyon16 Grand Challenge and Hubei Cancer Hospital. In our experiments, the average AUC was 0.95 in the test set for pixel-level detection.

## Introduction

In the past 100 years, pathologists have used microscopy to observe glass slides for clinical and pharmaceutical research, and more importantly, for providing definitive disease diagnoses to guide patient treatment and management decisions [1]. With the rapid development of whole-slide digital scanners for histopathology, computer-assisted digital pathology image analysis has increasingly attracted clinical attention [2]. In this rapidly growing field of digital pathology, computer-assisted image analysis systems have been confirmed to help pathologists diagnose tumors and cancer subtypes. In clinical practice, accurately distinguishing regions (normal and tumor) in digital pathology images is an important task that helps pathologists perform basic examinations and complement their opinion [3]. Thus, the workload of

**Data Availability Statement:** The data are held in a public repository, https://camelyon17.grand-challenge.org/Data/.

**Funding:** This work was supported by research grants 81300042 (to JY) from the National Natural

Science Foundation of China, [2014]41 (to JY) from the "Training Project for Young and Middle-aged Medical Talents" from the Wuhan Municipal Health Commission of China and WJ2019H124 (to JY) Health commission of Hubei Province scientific research project.

**Competing interests:** The authors have declared that no competing interests exist.

pathologists would be greatly reduced without any loss in sensitivity at the patient level. Pathologists can focus on making more complex and detailed diagnoses to ultimately provide more accurate results [4, 5].

Whole-slide digital scanners have become more prevalent in clinical hospitals and make it easier to digitize, store, share, visualize and analyze histopathology slides. Moreover, as one of the newest forms of "big data", whole-slide images (WSIs) in histopathology are constantly being produced every day. Typically, each WSI could have a full spatial resolution of $80K \times 80K$ pixels and is approximately 2 GB in compressed storage size at 40× magnification. This high volume of data requires the development of a fast and effective processing pipeline for analyzing digital image data.

In recent years, there has been increasing interest in developing computer-assisted image analysis methods in pathology. A variety of competitions have emerged to promote intelligent algorithm research on digital tumor histopathology. The early competition task was to perform cell segmentation and image-related feature extraction. The tasks are the classification and grading of more complex whole-slide pathological images.

The classification and grading of pathological images is the last step in the automatic analysis of pathological sections, and it is also a crucial step. In recent years, with the powerful tool of deep learning, researchers have applied CNNs in various cancer detection tasks and achieved good results. The champion team of Camelyon16, Wang [6], obtained an area under the receiver operating characteristic curve (AUC) of 0.925 for WSI classification using GoogLeNet and random forest classifier with feature engineering. Cruz-Roa [7] proposed HASHI based on a patch-based classifier with a 2-layer CNN, probability gradient from a heatmap, and Quasi-Monte Carlo sampling for WSI. The adaptive sampling algorithm used in this paper is derived from this method.

Han [8] proposed a multiclassification task to identify subordinate classes of breast cancer that uses a combination model of CNNs to analyze breast cancer histopathological images from the BreaKHis dataset. Valkonen [9] extracted a large number of quantitative descriptors of image texture, spatial structure, and distribution of nuclei and applied a random forest model to output confidence values indicating the likelihood of cancer cells. Xu [10] used a pre-trained AlexNet to extract the features of input patches and trained a linear SVM for segmentation in the MICCAI brain tumor challenge. Wan [11] constructed combinations of feature sets, including pixel-, object-, and semantic-level features derived from CNN, and utilized multiple SVM classifiers to determine breast cancer grades. Bayramoglu [12] proposed a multitask CNN to predict both malignancy and image magnification levels simultaneously to improve performance on the BreaKHis dataset. Alsubaie [13] proposed a deep CNN under multi-resolution to perform lung adenocarcinoma pattern classification. Sirinukunwattana [14] presented a segmentation performance comparison of 10 different network architectures for histology image classification problems.

In the BACH challenge of ICIAR 2018, one of the tasks consisted of performing pixel-wise labeling of clinical hematoxylin-eosin-stained histopathological WSIs in four classes. Many new methods for the automatic classification of breast cancer biopsies were proposed, and CNN dominated the challenge [15]. In [16], a fully convolutional network based on DenseNet [17] was proposed for performing pixel-wise labeling of WSIs. In [18], a two-stage patch-based approach was proposed, which consisted of an autoencoder to extract image features and an image-wise CNN to perform the classification of the whole image. In [19], an ensemble of four modified Inception-V3 models was proposed for increasing the generalization capability of different networks trained on random subsets of training data. For WSI, a sliding window was used to uniformly extract patches, and a refined heatmap using ResNet-34 was used to reduce potential misclassifications. [20] used an ImageNet pre-trained on DenseNet-161 for the

segmentation of WSIs. [21] used an encoder-decoder network. The encoder is composed of five convolutional processing blocks that integrate dense skip connections, group and dilated convolutions, and a self-attention mechanism following SENet [22], and the decoder follows the U-Net [23] structure with skip connections between the down-sample and up-sample.

Li [24] proposed a neural conditional random field (NCRF) deep learning framework to detect cancer metastasis in WSIs. NCRF considers 9 spatially adjacent patches through a fully connected CRF, which is incorporated on top of a CNN feature extractor based on ResNet. Tokunaga [25] aggregated three expert CNNs based on U-Net by using three different magnification images and used a modified Xception [26] model to adaptively change the weight of each expert network depending on the input image. Li [27] developed a graph convolutional neural network to learn global topological representations of WSI for providing more accurate survival risk predictions. Wang [28] proposed a recalibrated multi-instance network for adaptively aggregating the patch information to image-level prediction of whole slide gastric image, which improved image-level classification accuracy by assigning different weights to each instance. Sun [29] applied U-Net to extract pixel-level features and adopt multiple classic fine-tuned CNN to obtain patch-level features, then jointed them by a hierarchical conditional random field method to localize abnormal (cancer) regions in gastric histopathology images.

In recent years, deep learning in solving image classification tasks, such as classification on ImageNet, has been greatly successful. Deep convolutional neural network (DCNN) models have been reported to surpass human performance. These models are typically used to process relatively small-sized natural images ($200 \times 200$ pixels), but WSI is over hundreds of times the size of a natural image. Therefore, most pathology image analysis methods take a patch-based classification approach that first segments a large image into small patches and then classifies each patch. This piecemeal approach has limited their analysis to small regions of interest (ROIs) within the larger WSI. Thus, the overall size of the neural network can be allocated in the GPU memory. The issue associated with this approach is the need to use a sampling mechanism to traverse the entire pathology image. Dense uniform or regular sampling is one of the practical options, but the efficiency is not high. Even if there is no overlap between sample patches, a full detection process for the WSI is required to extract tens or hundreds of thousands of patches. In contrast, the adaptive sampling method is a more effective strategy for dealing with WSIs because it adaptively chooses regions with high uncertainty of a tissue patch being cancerous or not. For regions wherein the predictor has a greater uncertainty about cancer and normal tissue classification, more patch samples will be classified to improve the confidence of the adaptive sampling method for those regions of ambiguity [7].

To establish a complete WSI processing pipeline, there are still some issues to discuss after the patch-based classification and adaptive sampling mechanism are selected, such as, how to develop an efficient and accurate classifier, what is the more appropriate convergence condition of the iterative sampling process, how to use images under a wider range of magnifications?

Here, we present a deep learning-based approach for the identification of tumor metastasis on WSIs from the Camelyon16 dataset [30]. In summary, the main contributions of our study are as follows:

- Based on High-throughput Adaptive Sampling for Whole-slide Histopathology Image analysis (HASHI) [7], we propose an improved adaptive sampling method with superpixel segmentation and introduce a new heuristic factor, with local sampling density as the convergence condition of iterations to improve the detection effect of multiple instances.

- We develop the Double Magnification Combination (DMC) classifier, which is a modified DenseNet-40 to make patch-level predictions to discriminate tumor patches from normal

patches. The lightweight network use 20× and 40× magnification images with only 0.3 million parameters, and uses the large-margin Gaussian Mixture (L-GM) loss function [31] to improve the generalization performance.

- In postprocessing, we train a CNN model with 4 convolutional layers to regulate the patch-level predictions based on the predictions of adjacent sampling points.

The source code for our approach has been made publicly available at https://gitee.com/w3STeam/Pathological-images and https://github.com/JustinRuan/Pathological-images.

## Materials and methods

Our tumor metastasis detection framework consists of a patch-based classifier, an improved adaptive sampling method, and a postprocessing filter. The complete pipeline is divided into two stages, namely, the sampling stage and the postprocessing stage, as shown in Fig 1.

### Patch extraction and preprocessing

Our model was trained with the Camelyon16 dataset, which consists of 400 WSIs total, split into 270 WSIs for training and 130 WSIs for testing. Here, the extraction of patches was divided into two cases: extraction for generating a training set and extraction in adaptive sampling for detection.

To focus our training data set on regions of the slide most likely to contain tumor metastasis, we first identified tissue within the WSI and excluded background white space. There are many methods based on threshold segmentation, such as [6, 9, 11]. We adopt a fixed-level threshold segmentation method in the HSV color space to exclude the obvious background region. The final mask images were generated by combining the masks from the S and V

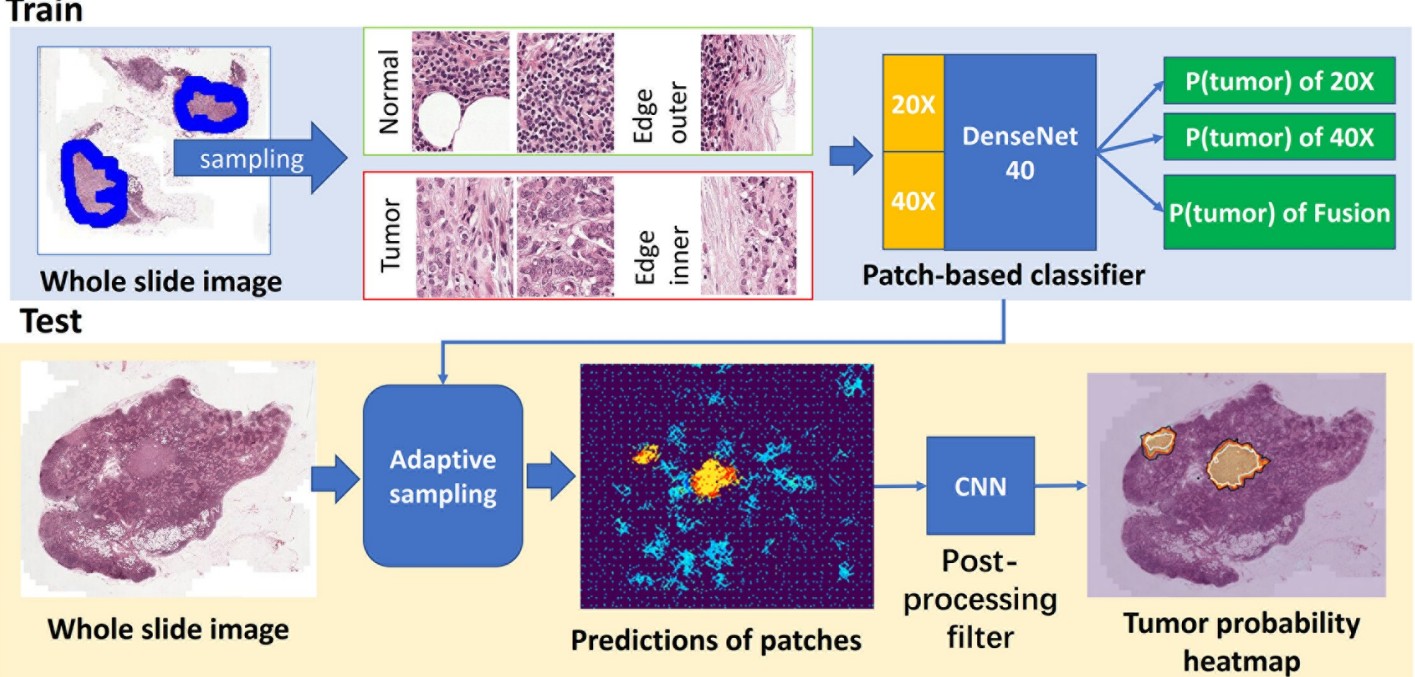

**Fig 1. An overview of our proposed workflow.**

channels. The constraint of effective coverage is that the threshold of the V channel is between 0.2 and 0.8, and the threshold of the S channel is greater than 0.1.

According to the detection results and pathologist's annotation, we extracted four types of patches: normal, tumor, edge inner, and edge outer. The labeling of a patch is determined by the proportion of the tumor area within the patch. When the proportion of tumor area is less than 50%, this patch is normal (Label 0); otherwise, it is tumor (Label 1). For the labeling under the combination of double magnifications, we adopted the "or" logic here. Here We used morphological methods to extract edge regions and increased the number of training samples at the edge of annotations to improve the performance of the classifier. Because the patches at the edge of annotations are usually transitional regions from tumor tissue to normal tissue, most of the "hard examples" are concentrated in these positions.

At a sampling point, we simultaneously extracted two patches under 20× and 40× magnifications, and the size of the patches was 256×256, as shown in Fig 2. Preprocessing and normalization were not applied to these saved patches to preserve the inherent fluctuation characteristics caused by staining. These are also what the classifier needs to fit. We obtained a total of 1,694,228 patches, with 1,336,704 labeled as normal, 230,966 labeled as tumor, 57,384

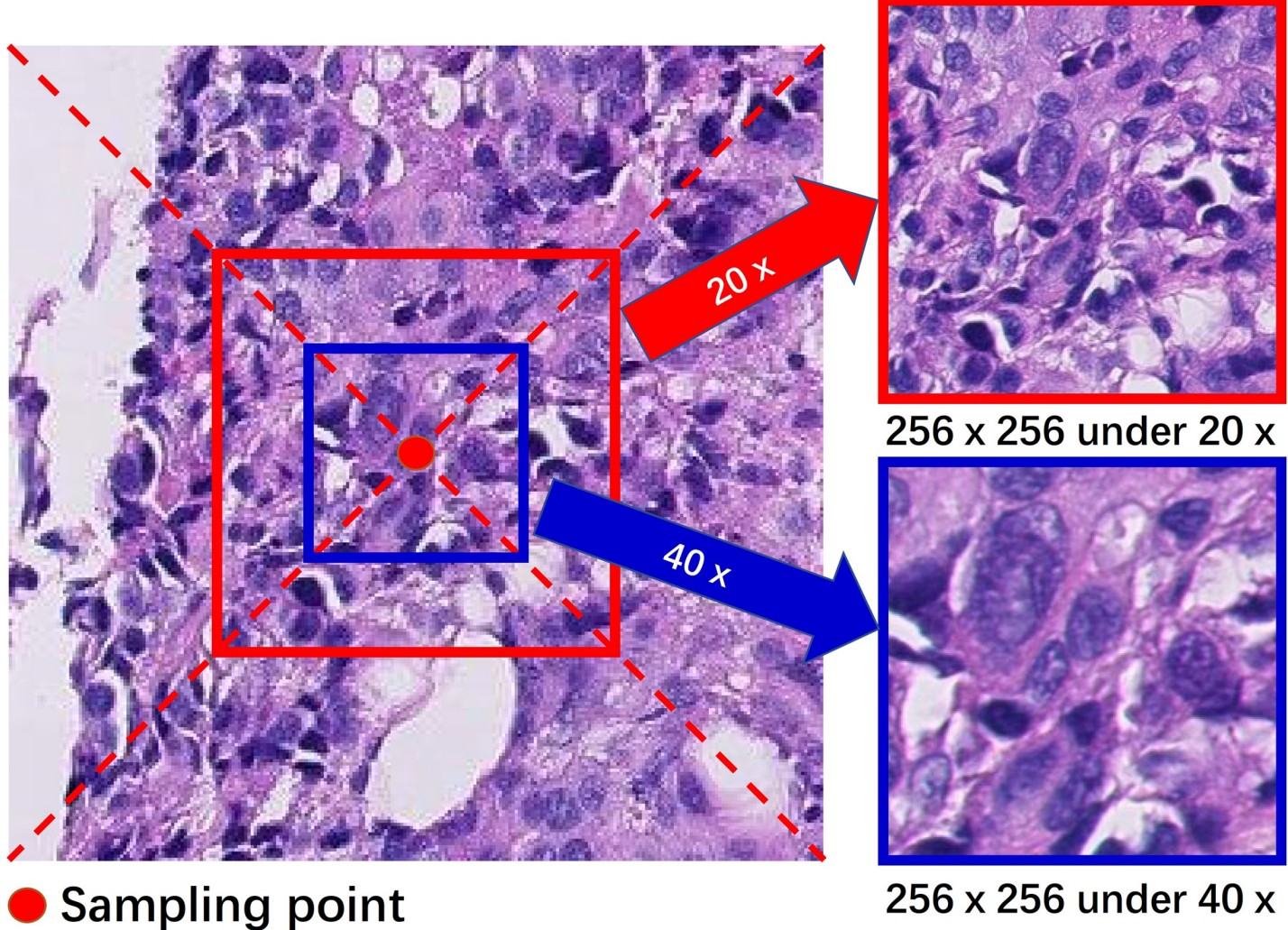

**Fig 2. Extract two magnification patches at a sampling point.**

labeled as edge inner, and 69,174 labeled as edge outer. It is worth noting that we balanced the number of positive and negative samples in a WSI. The sampling interval in the normal regions is larger than that in the tumor region. In this way, the number of negative samples in a WSI does not exceed 5~6 times the number of positive samples. Then, we constructed several balanced training sample sets (1:1) through random sampling for improving the performance of the patch-based classifier.

For sampling and predicting, we first used the threshold segmentation method mentioned above to calculate the effective region of the slide. Then, we directly extracted the 20× and 40× patches at pseudo-random sampling points and input them into the patch-based classifier. As in training, these extracted patches do not require any preprocessing or normalization.

## Architecture of the patch-based classifier

To explore the appropriate classifier structure, We chose nine classic pre-trained ImageNet networks to test the patch-based classifier under three magnifications. We replaced the original top layer with a new one to connect each feature extraction part, which consists of a Global Average Pool (GAP) and two fully-connected (FC) layers. We used the prepared patches on three different magnifications to fine-tuning the top layer of each transfer model and tested the accuracy of these models. All testing results of transfer-learning are shown in S1 Table in S1 Appendix. According to the results of transfer-learning, the DenseNet family has the best feature extraction performance for pathological image blocks. The patches under 20× have the best distinguishable characteristics that can be extracted by CNN, as a result of the balance of the texture details and texture range in view. Although the 10× patches have a larger field of view, they are down-sampled to the same size resulting in the loss of texture and degradation of classification performance. Compared to 20×, the classifiers under 10× are a little worse. Under 40×, the field of view in a patch becomes very small in a patch. When the patches are extracted from the transitional zone from tumor to normal near the edge of annotations, these patches under 40× are no significant and typical texture features, and even look the same as the patches in normal regions. So, it is difficult to train a better classifier under this magnification individually. On the other hand, the patch-based classifier calculates a tumor feature based on an entire $256 \times 256$ image, and the calculated patch-level prediction is stored in a tumor feature map based on the central coordinate of this patch. Thus, with the same image size, the prediction under higher magnification can more accurately represent the tumor feature (probability) at the sampling point (the center of a patch). From the perspective of spatial location, we argue that the prediction of a patch with the same size under 40× is more accurately express the tumor feature at the center of the patch, and facilitate the generation of more detailed segmentation boundaries. Moreover, in the pixel segmentation experiment, the accuracy under the 20× and 40× magnifications alone is better than that under the 10× alone.

Pathologists usually check images by changing their magnification and scope in the WSI. Ways to use images under a wider range of magnifications are worth studying. Inspired by this, we investigated the patch-based classifier with multiple magnifications. We finally chose the combination of patches under 20× and 40× as inputs at the same sampling coordinates.

Our patch-based classifier was derived from DenseNet40 (= 3x6x2+2x1+1+1 = 40). The network consists of three dense blocks defined in DenseNet. Each block consists of 6 dense layers that each contains two convolution layers. Between two adjacent dense blocks, there is a translation layer that consists of one convolution layer. And only two transport layers are used here. The network also contains a convolutional layer at the input and a fully connected layer at the top. We called it the Double Magnification Combination (DMC) patched-based classifier. The network contains two inputs and three outputs, as shown in Fig 3. Our modified

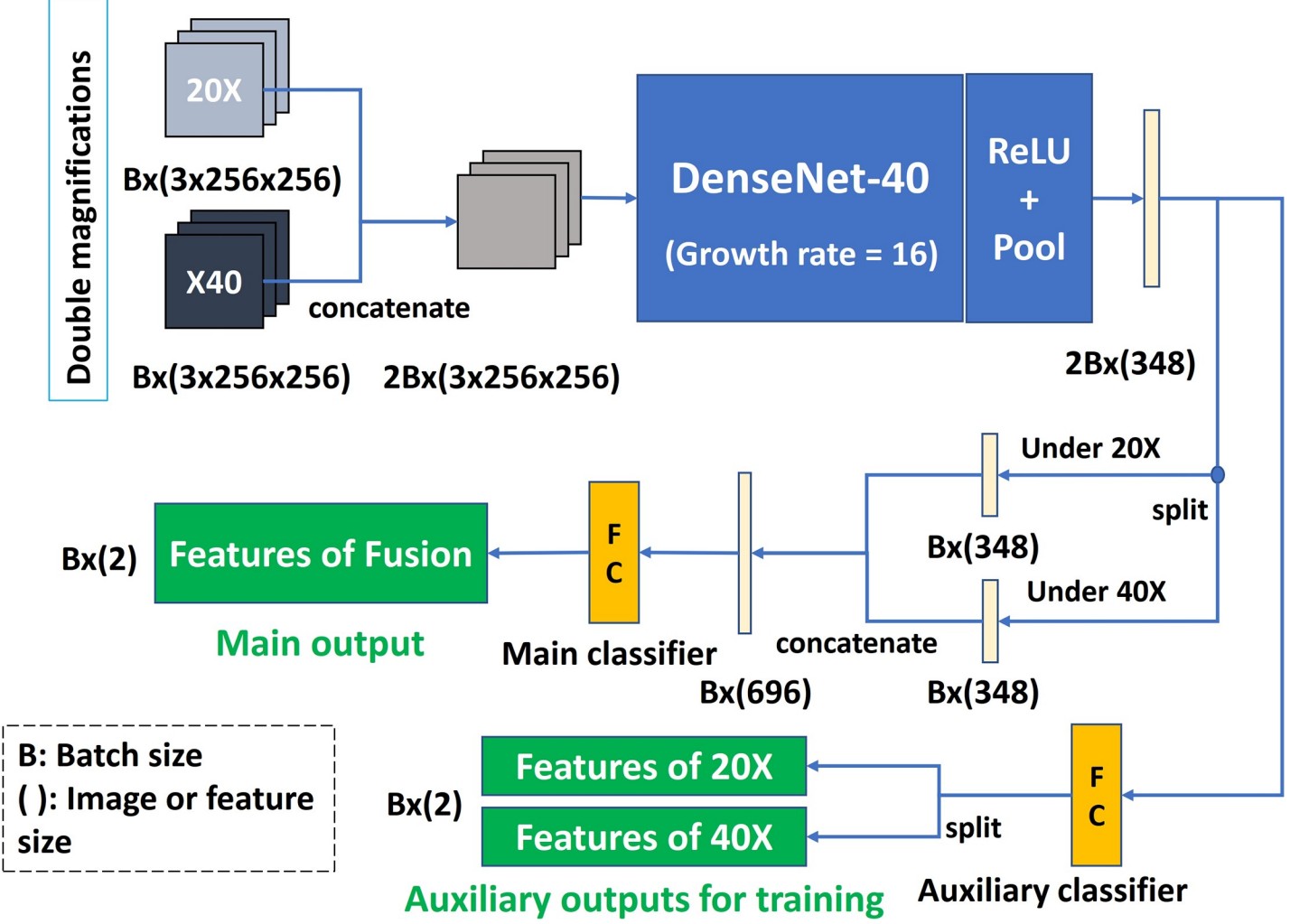

**Fig 3. DMC patch-based classifier architecture.**

network has only 0.3 million parameters. The growth rate ('k' in [17]) is set to16 to reduce the parameters of the model by using very narrow layers, at the same time, keeping up with the performance of our patch-based classifier.

To improve accuracy and generalization, the same network was used to process two types of patches under 20× and 40×. The two patches at the corresponding sampling points are put into the same batch in a fixed order. Each three-channel image generated separately a 384-dimensional feature by DenseNet. Because each sampling point includes two input images (patches), we stitch the two-output feature of the same sample point into a 696-dimensional vector and consider it as the fusion feature of the sampling point.

After the output features are continuously stitched and split, the main classifier uses double-magnifications patches at the same sampling point for prediction, while the outputs of the auxiliary classifier under single magnification are only used as additional outputs for training. We used the regularization term to drive the output features of the auxiliary classifier under 20× and 40× to obey the same Gaussian mixture distribution. The final outputs here are 2-dimensional features, not probabilities. The "SoftMax" layer is not included in the network because our adaptive sampling algorithm mainly uses feature space.

## Training

We used the training data set with two magnifications to train our patch-based classifier. To encourage the robustness and generalization of our network, we used two loss functions with three prediction outputs ($\tilde{y}_{20}$, $\tilde{y}_{40}$, and $\tilde{y}$). Among them, $\tilde{y}_{20}$ corresponds to the predictions of patches only under 20×, $\tilde{y}_{40}$ corresponds to the predictions under 40×, and $\tilde{y}$ refers to the prediction of the fusion feature. Here, $y$ refers to the ground truth under double magnification, $y_{20}$ and $y_{40}$ and so forth.

The first loss function uses only the cross-entropy loss between $\tilde{y}$ and $y$. Here, $\mathcal{L}_{CE}$ is the cross-entropy loss function.

$$loss_1 = \mathcal{L}_{CE}(\tilde{y}, y) \tag{1}$$

Through $loss_1$, the accuracy of the main classifier is improved, and the fusion features under double magnification can be better discovered and utilized. After $loss_1$ is backpropagated, the temporarily generated graph used to compute the gradient of the network needs to be preserved. Next, we perform the backpropagation of the second loss in (2), which consists of the average of cross-entropy loss under single magnification and a regularization term based on the large-margin Gaussian Mixture (L-GM) loss [31]. The regularization term simultaneously drives the deep model to generate the same Gaussian mixture-distributed features under two different magnifications. Because of its use, the generalization capability of the trained model is improved.

$$loss_2 = 0.5 * [\mathcal{L}_{CE}(\tilde{y}_{20}, y_{20}) + \mathcal{L}_{CE}(\tilde{y}_{40}, y_{40})] \tag{2}$$

$$+w * \mathcal{L}_{GM}([\tilde{y}_{20} \quad \tilde{y}_{40}], [y_{20} \quad y_{40}])$$

Here, the coefficient of the first term is 0.5 indicates that the two cross-entropy losses at both magnifications have the same weight, that is, the classification error is reflected in their average under both magnifications. L-GM loss includes a nonnegative hyper-parameter $\alpha$ for controlling the expected margin between two classes in the training set. And its default value is 1.0 in [31]. We followed this setting. And $w$ is the weight of the regularization term $\mathcal{L}_{GM}$, which is 0.001 by default. Without data amplification, we trained the whole network for 40 epochs with a learning rate of $1 \times 10^{-3}$ and 40 epochs with a learning rate of $1 \times 10^{-4}$. The training was performed using PyTorch.

## Improved adaptive sampling method

HASHI [7] provided a feasible solution for slide-level scanning and prediction on WSIs. After training a patch-based CNN classifier, HASHI extracts patches from the WSI using Quasi-Monte Carlo sampling and predicts the tumor probabilities of these patches. These predictions are used to build an interpolated probability map, which is used to identify suspicious regions for further sampling. The newly sampled patches are used to produce an improved probability map estimation. The iterative process does not end until the limit of the maximum iterations is reached, and the final probability map is produced.

Our inspiration comes mainly from HASHI, and the main objective of our improved method is to try to optimize the following aspects.

- Change in the algorithmic structure. At the initial sampling, a regular sampling process based on superpixel segmentation is added. After an adaptive sampling of the full slide, an iterative process based on the partial superpixels is added.

- Change in the selection conditions of the sampling points. The original algorithm sorts by the gradient of the probability map, then select the coordinates within the larger half for

sampling. Our algorithm uses a cluster-based heuristic factor to select sampling points based on feature gradients.

- Change in the convergence condition. Compared to the maximum number of iterations in the predecessor, we introduced a new statistical factor, local sampling density, to judge whether the iterations should be terminated.

**Regular sampling during initialization.** For the detection of whole-slide images, the general standard multiple instance assumption needs to be considered. In HASHI, sampling points are more likely to be enriched near a larger area of the tumor region. A larger area has a longer edge, which corresponds to a tumor probability gradient change. The adaptive sampling algorithm preferentially detects these locations where the tumor probability gradient changes are large. In contrast, small tumor areas do not have significant tumor gradient changes, which may result in under-sampling in certain suspicious regions. Also, when the areas of the tumor regions in the WSI are small, the iterative sampling process may untimely terminate due to the limit of the number of iterations. To avoid this, we have to increase the number of maximum iterations or the number of samples per iteration, which means that it is necessary to guarantee a minimum number of samples.

We extracted the thumbnail $I$ of a WSI $X$ under 1.25× magnification (Level 5) and separated it using the SLIC [32] algorithm (compactness = 20). The boundaries $B$ of the segmented superpixel regions $S$ were extracted. Then, we performed regular sampling at uniform spatial intervals on the boundaries of $S$. Here, the number of superpixels $S$ is proportional to the area of the WSI. The area of each superpixel $S$ is approximately 1000 pixels under 1.25×, which is equivalent to the area of four 256×256 patches under 20×. In this way, a set of center coordinates $C_R$ of patches is obtained and used to generate the first gradient map in the feature space. Unlike the original algorithm, our gradient maps are based on features rather than probabilities. Because the feature space has a larger dynamic range than the probability space, more edge details are obtained.

**Adaptive sampling within full scope.** The first stage strategy extracts the random coordinates $C_A$ of $N_A$ sampling points using Quasi-Monte Carlo sampling and merges them with the previous regular coordinates $C_R$. Here, we chose Halton sequences [33–35] to generate the coordinates of the sampling points. The patches were extracted in pairs under double magnification and put into our two-input classifier. The predications of these patches produced an initial coarse estimation of a linear interpolated feature map $M_{feat}$. Then, we generated a gradient map $M_{grad}$ of the estimated feature map using the Sobel algorithm. Next, Mini Batch K-Means clustering [36] was applied on $M_{grad}$ to partition the feature gradients into two clusters. At least one of the cluster centers $\mu_0$ of gradients is close to zero, which corresponds to flat regions in the gradient map (typical tumor or normal regions in the WSI). If another cluster center $\mu_1$ is also close to zero, no significant edges are found in the current $M_{grad}$. The edges of $M_{grad}$ correspond to the regions with large gradient change, that is, the uncertain or suspected tumor regions. The cluster center $\mu_1$ corresponding to the possible gradient edge should have a larger value.

Here, we introduced a new heuristic factor $f_{grad}$ to determine whether the edges of $M_{grad}$ are found, as shown in (3). The value range of $f_{grad}$ is from 0 to 0.3.

$$f_{grad} = \min(0.5 * (\mu_0 + \mu_1), 0.3) \tag{3}$$

If $f_{grad}$ is greater than the threshold $T_{grad}$ (0.03), our adaptive sampling algorithm only

focuses on the position where the feature gradient is greater than $f_{grad}$. Otherwise, the sampling algorithm continues to pseudorandomly search sampling coordinates in the full image.

In summary, the generation algorithm of the sampling points is divided into three cases. The first case is to randomly generate sampling points using a Halton sequence in the full scope of a WSI. The second case is that none of the uncertain or suspected tumor regions have been found in $M_{grad}$. The generation algorithm pseudorandomly searches sampling coordinates and preferentially selects sampling coordinates with higher gradients in its 16×16 neighborhood under 1.25×. This is equivalent to reducing the size of the gradient map $M_{grad}$ to the original one-sixteenth size through maximum pooling. The generation algorithm searches for sampling coordinates based on this reduced gradient map to enhance its overall discovery capabilities. The third case is that the generation algorithm focuses on searching for suspicious regions when $f_{grad}$ is greater than $T_{grad}$. At this time, only the coordinates whose corresponding gradient is greater than $f_{grad}$ will be selected. If the algorithm cannot find enough sample points that satisfy the constraint at once, then it will look again in the neighborhood of the sample points just selected.

Through iterative adaptive sampling, $M_{grad}$ is continuously refined until the convergence condition of the first sampling stage is reached. Here, we introduced a new statistical factor, local sampling density $\rho$, which is the number of previous sampling points in the neighborhood of a new sampling point. There are two forms of neighborhoods here. One of them, $\rho_{dt}$, is to define the range of the neighborhood by distance. The other, $\rho_{sp}$, is defined by belonging to the same superpixel, which is only used in the second stage.

$$\rho_{dt}(c_i) = \sum I(\dot{c}_j | \|\dot{c}_j - c_i\| < \varepsilon) \tag{4}$$

$$\rho_{sp}(c_i) = \sum I(\dot{c}_j | c_i, \dot{c}_j \in s_k), s_k \in S \tag{5}$$

Here, $c_i$ is the coordinate of the sampling point $i$ in the current iteration. $\dot{c}_j$ is the coordinate of $j$ in previous iterations. The symbol $\|\cdot\|$ is a distance function, such as Chebyshev Distance. $\varepsilon$ indicates the size of the neighborhood of sampling points. $s_k$ is a superpixel in the segmentation results $S$.

When the average of $\rho_{dt}$ of the current iteration is greater than the threshold $T_\rho$, the sampling process will enter the second stage. An adaptive sampling process similar to the first stage is performed in a part of the superpixels. We usually set $T_\rho$ to 1 or 2, since setting it to a larger value has little effect on the results but takes more time. See the S1 Appendix for this algorithm pseudocode.

**Adaptive sampling within enabled superpixels.** Once the average of $\rho_{dt}$ reaches the threshold $T_\rho$, the adaptive sampling algorithm will further explore regions where the sampling density is low but the tumor probability is high. Therefore, we excluded part of a WSI based on the superpixels obtained earlier. We counted the number of sample points contained in each superpixel, that is, the local sampling density $\rho_{sp}$. Based on the interpolation feature map and the gradient map, the two maximums $\hat{f}_{max}(s_i)$ and $\hat{g}_{max}(s_i)$ in the $i$th superpixel were also calculated. Here, we used three thresholds $T_\rho$, $T_f^{sp}$ and $T_g^{sp}$ to determine which regions need further inspection.

$$S_{enable} = (s_i | \rho_{sp}(s_i) < T_\rho \wedge \hat{f}_{max}(s_i) > T_f^{sp} \wedge \hat{g}_{max}(s_i) > T_g^{sp}) \tag{6}$$

Because the output of our binary patch-based classifier is the tumor feature of a patch, if we use the Sigmoid function to regress a feature into a tumor probability, the feature value -1 corresponds to the tumor probability of 27%. Here the threshold $T_f^{sp}$ represents the lower limit of

the tumor feature in a superpixel, generally set to -1. When there is a feature larger than $T_f^{sp}$ in a superpixel, it means that there is a point with a tumor probability greater than 27% inside. In the next iteration, such superpixels will be further explored. $T_g^{sp}$ is 0.1; this gradient threshold constrains the regions in $S_{enable}$ from being too flat. Because such flat regions are generally far from the boundaries of the tumor, too much sampling does not contribute much. $\rho_{sp}(s_i)$ indicates whether the superpixel $s_i$ is fully sampled.

When $S_{enable}$ is updated, the adaptive sampling process is executed again until the average of $\rho_{dt}$ reaches $T_\rho$. By iteratively updating $S_{enable}$ and sampling, $S_{enable}$ finally becomes an empty set, and the entire algorithm will end. It should be noted that if the two thresholds $T_f^{sp}$ and $T_g^{sp}$ are sufficiently small, such as -3 and 0, the entire sampling process will degenerate into uniform sampling.

Fig 4A shows the 112 (purple) sampling points generated in the first round of the adaptive sampling, and 12 of which were generated by regular sampling during initialization. In Fig 4B–4D, the next three rounds of sampling are shown here, and the 4th round reached the termination condition $T_\rho$. The sampling points were densely generated where the predicted tumor probability exceeds 0.3 and they had a larger gradient in the estimated feature space.

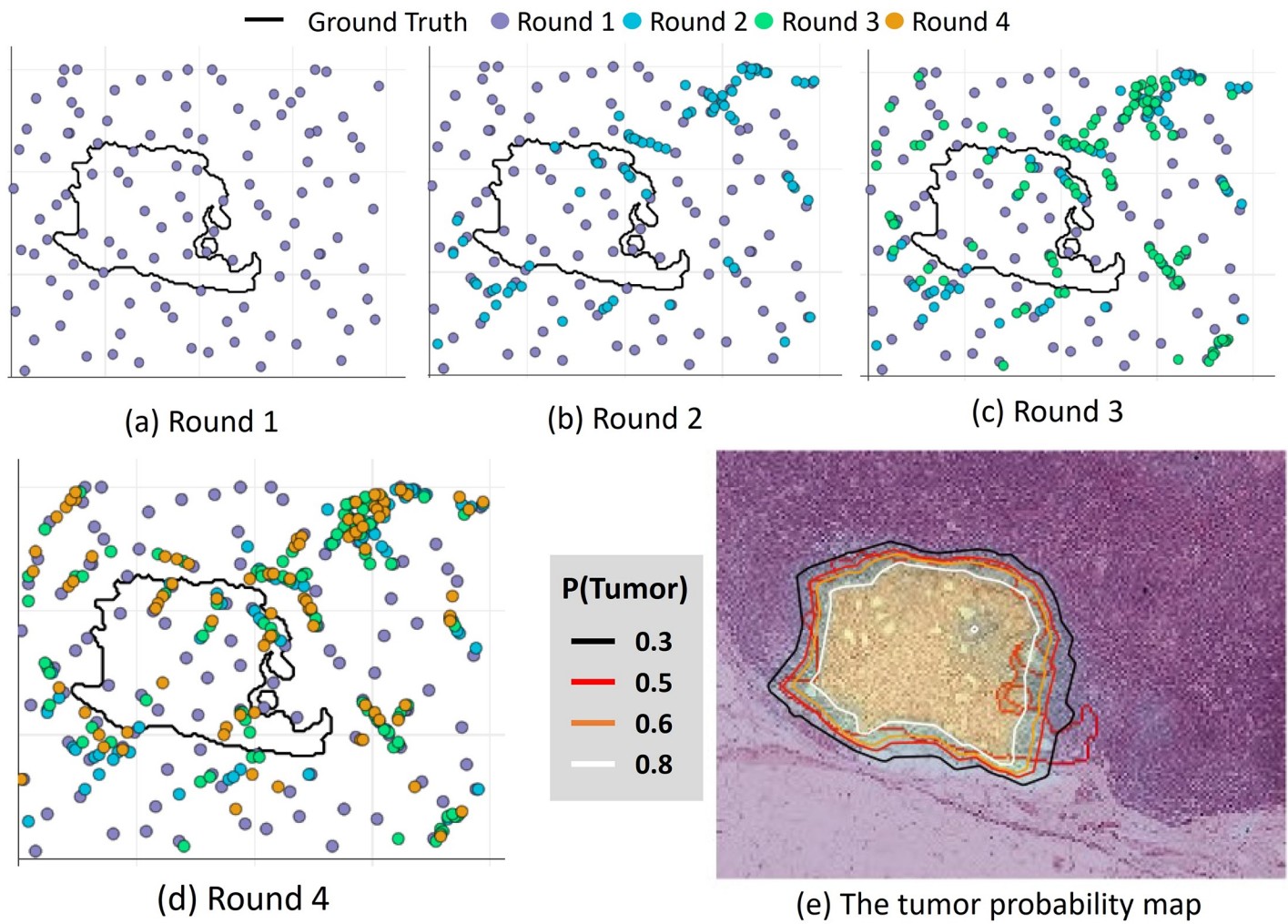

**Fig 4. The process of the adaptive sampling.**

Fig 4E shows the contour lines of the tumor probability map with different colors. The red line indicated the ground truth.

Please refer to the S1 Appendix for the detailed process of the sampling algorithm.

```
Algorithm 1: Adaptive gradient-based sampling
Input:
    M: CNN-trained model
    X: WSI
    T: maximum iterations
    N_A: number of sample points extracted
    A: area of each superpixel
    d: spaced intervals of sampling
```

$T_{grad}$, $T_{\rho}$, $T_f^{sp}$ and $T_g^{sp}$: thresholds
$M_{grad}$, $f_{grad}$, $\mathcal{H}, \ldots \leftarrow \phi$

```
    S, C_R ← regular sampling based on superpixels (X, A, d)
    S_enable = S
    For i = 1 to in T do:
        C_A ← sampling point generation (N_A, M_grad, f_grad, T_grad, S_enable,...)
```

$$C = \begin{cases} C_R \cup C_A, i=1 \\ C_A, \qquad i>1 \end{cases}$$

```
        Predictions F ← patch classification (M, C)
        M_feat ← feature map interpolation (F, C)
        M_grad ← feature gradient (M_feat)
```
$\mu_0$, $\mu_1 \leftarrow$ clustering ($M_{grad}$)
$f_{grad} = \min(0.5^*(\mu_0+\mu_1), 0.3)$
$avg_{\rho_{dt}} \leftarrow$ average local sampling density within the neighborhood
$(\mathcal{H}, C)$
$\mathcal{H} \leftarrow ((c_i, f_i) | c_i \in C, f_i = \mathcal{F}(c_i)) \cup \mathcal{H}$
If $avg_{\rho_{dt}} > T_{\rho}$:
 $\rho_{sp} \leftarrow$ local sampling density within superpixels ($\mathcal{H}, S$)
 $S_{enable} \leftarrow$ update enabled regions ($S, \rho_{sp}, T_{\rho}, T_f^{sp}, T_g^{sp}$)
 If $S_{enable}$ is $\phi$:
 **Return** $\mathcal{H}$

Here, $C_R$ refers to the set of center coordinates of patches, which are obtained by regular sampling based on superpixel segmentation. $C_A$ refers to the set of center coordinates of patches, which are obtained by random sampling (Quasi-Monte Carlo) process. We only returned the coordinates $c_i$ and predictions $f_i$ (1-dim feature) of each sample point in all iterations. Next, the postprocessing generates a heat map of tumor probability.

In our experiments, a slide was required to extract nearly 59000 patches of size 256×256 on average using uncovered regular sampling under 20×. Our sampling method with parameters $T_{\rho} = 1$ and $N_A = 2000$ only needed to extract an average of 7400 patches, which is only 1/8 of the workload of the uncovered regular sampling.

## Postprocessing

In postprocessing, each obtained prediction is adjusted based on the predictions of its neighboring sampling points. A CNN model with 4 convolutional layers was trained to regulate the patch-level predictions under 1.25×, as shown in Table 1. We can think of this as adaptive filtering of patch-level features, so we also called it a slide filter. The input of the slide filter is a 64×64 single-channel matrix centered at each sample point, which includes the feature of its adjacent sampling points. If the sample point is in a tumor region, it is a tumor/positive sample and labeled as 1; otherwise, it is a normal/negative sample and labeled as 0. Corresponding to 20×, the size of the input matrix is 1024×1024 pixels, and the area is equivalent to the 16 non-overlapping patches used in the patch-based classifier.

**Table 1. The slide filter (CNN model) in postprocessing.**

| Layer (type) | Output Shape | Param |
|---|---|---|
| Conv2d+ReLU | [32, 64, 64] | 320 |
| Conv2d+ReLU+MaxPool2d | [32, 32, 32] | 9,248 |
| Conv2d+ReLU+MaxPool2d | [48, 16, 16] | 13,872 |
| Conv2d+ReLU+MaxPool2d | [64, 8, 8] | 27,712 |
| AvgPool2d | [64, 1, 1] | 0 |
| Linear | [2] | 130 |
| Total params | | 51,282 |

According to the pathologist's annotations and the obtained predictions, we generated a training set of 188360 balanced samples (the ratio of normal to tumor samples is 1:1). The loss function used in training consists of two parts: cross-entropy loss and L-GM loss. The network was trained on patches of shape = 64×64 pixels, with batches of size = 200, and weight of L-GM loss = 0.001.

The corrected patch-level predictions are generated by a weighted average with the new patch-level predictions and the original prediction. Then, according to the corrected predictions and sampling coordinates, a tumor probability heat map is generated by the Sigmoid function and linear interpolation. Here, we did not use fully-conv (FC) net to directly generate a heatmap under 1.25×, because this required higher hardware requirements (GPU memory capacity).

We used the tumor probability heatmap to compute the evaluation for each WSI. In Fig 5, the contour lines of the probability maps are shown in (A). (C) is a partial enlargement of the lower right corner of (A). Here, the contours with different colors correspond to different probabilities. (B) and (D) show the predictions corresponding to the left side using our adaptive sampling method. The yellow regions in (B) and (D) indicate the ground truth. We gave more examples in the S1 Appendix.

## Results & discussion

This paper used the H&E-stained WSIs of the Camelyon16 challenge, which is aimed at detecting metastasis on the WSIs of lymph node sections [6], and 40 H&E-stained WSIs provided by Hubei Cancer Hospital (HCH). We used these datasets to detect tumor regions. The tumor regions in the HCH dataset are generally large and typical, as shown in Fig 6. In the two subplots, the blue regions are the marked tumor regions. In Fig 6B, a green region is the excluded region. In terms of the number of tumor regions per slide, the test set samples in Camelyon16 contain an average of 33 compared to an average of 11 for the HCH samples.

### Evaluating the patch-based classifier

In this section, we mainly evaluate the performance of the patch-based classifier. To compare the classification performance of different networks, we used the prepared dataset to evaluate their patch-level F1 scores, as shown in Table 2. Here, "40×" refers to the DenseNet-40 with a single input under 40×. "DMC" refers to our modified DenseNet-40 with two magnification inputs, "DMC 40×" refers to the auxiliary classifier only for 40×, "DMC 40×+20×" refers to the main classifier using fusion features under two magnifications. "L-GM" refers to the L-GM loss used in training. From the patch-level results of the classifier, the performance of the auxiliary classifier under 40× of DMC is similar to the single-input classifier under 40×, and that of the auxiliary classifier under 20× is better than the corresponding single-input classifier. When

**Fig 5. Tumor probability heatmap and predictions of sampling points.**

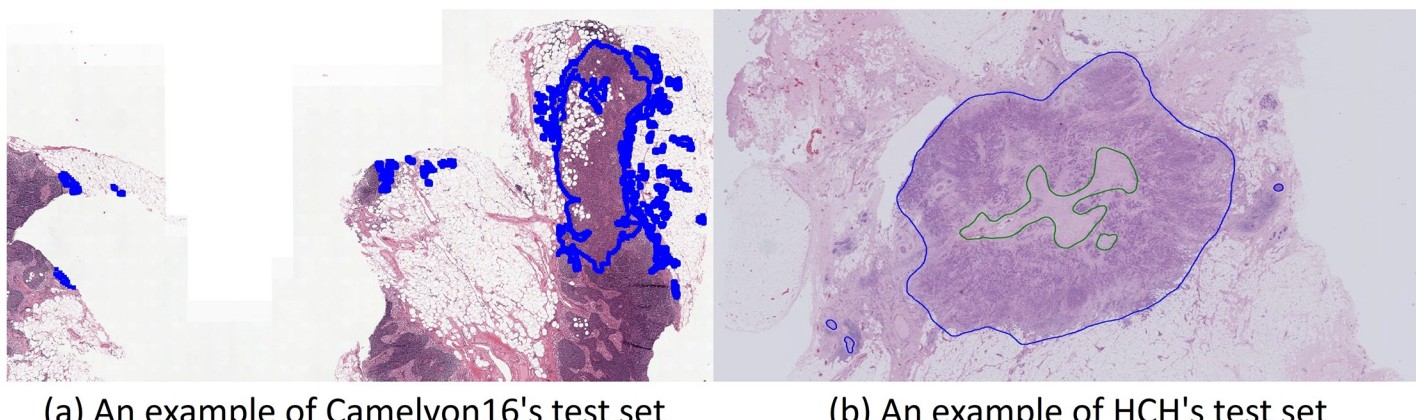

(a) An example of Camelyon16's test set

(b) An example of HCH's test set

**Fig 6. Examples of WSI in the two datasets.**

the fused features under both magnifications are used at the same time, the performance was improved by nearly 2~3%. Because a pair of patches overlap at the center point and the field of view is different, so the spatial attention mechanism was introduced. For the performance of patch-level detection, our experience is the use of L-GM loss in training has no significant effect on single-input or dual-input classifiers.

Next, we evaluated the performance of pixel-level detection, and this evaluation more tested the generalization capability of patch-based classifiers. Using the pathologist's annotation as the ground truth, ROC analysis at the pixel level heat map was performed, and the measures used for comparing the algorithms were F1 score and area under the ROC curve (AUC). In Table 2, the pixel level detection results all used our proposed adaptive sampling algorithm, and the parameter configuration for our model involved the threshold $T_\rho$ of the local sampling density ($T_\rho = 1$) with 2000 samples per iteration ($N_A = 2000$) and the area of each superpixel $A = 1000$ pixels with spatial intervals of regular sampling $d = 60$ pixels under 1.25× magnification. Besides, the pixels with tumor probability greater than 0.5 were considered positive in the heat map.

Regarding the F1 scores in Table 2, there is not much difference in accuracy between single-input or dual-input patch-based classifiers, but there is a significant difference in the results of pixel-level segmentation. In the pixel-level segmentation task, the F1 scores of each patch-based classifier are much lower than the scores of the classifier during training and testing of patches. Note that, the loss of the pixel-level segmentation task is not used to optimize the performance of patch-based classifiers; it represents the generalization performance of the classifier. This is because, although we extracted millions of patches from WSI for training these patch-based classifiers, the input images during our adaptive random sampling are almost impossible to be the same as those in the training set. In other words, the patches extracted during the adaptive sampling are more varied. Moreover, the prediction error at any sampling point has an impact on the accuracy of the segmentation boundary near it. The superposition effect brought by the sampling mechanism makes it possible to obtain correct results only when the robustness of the patch-based classifier is sufficient. Regarding the F1 score of pixel level, the performance of our adaptive sampling algorithm on DMC is nearly 20% higher than that of the classifiers with single magnification. This shows the advantages of the dual input structure.

From the detection results of the pixel level, L-GM loss is necessary for DCM. The use of L-GM loss increases the margin between the centers of the two classes, During the adaptive sampling process, more the features of sampling patches fall into this gap, resulting in the deterioration of the detection results. The contours of the probability of 0.5 at the heat map

**Table 2. The classifier detection performance.**

| Methodology | | Patch Level | | | | | | Pixel Level | | | |
|---|---|---|---|---|---|---|---|---|---|---|---|
| | | Train | | | Test | | | Train | | Test | |
| | | F1(Normal) | F1(Tumor) | F1(Avg) | F1(Normal) | F1(Tumor) | F1(Avg) | F1 | AUC | F1 | AUC |
| 40× | | 0.9546 | 0.9548 | 0.9547 | 0.9711 | 0.8827 | 0.9269 | 0.5170 | 0.9616 | 0.4595 | 0.9030 |
| 20× | | 0.9567 | 0.9570 | 0.9568 | 0.9701 | 0.8805 | 0.9253 | 0.5738 | 0.9286 | 0.5036 | 0.8571 |
| DMC | 40× | 0.9512 | 0.9514 | 0.9513 | 0.9711 | 0.8844 | 0.9277 | - | - | - | - |
| | 20× | 0.9712 | 0.9713 | 0.9712 | 0.9802 | 0.9242 | 0.9522 | - | - | - | - |
| | 40×+20× | **0.9737** | **0.9740** | **0.9738** | 0.9810 | 0.9275 | 0.9542 | 0.6007 | 0.8454 | 0.5518 | 0.8630 |
| DMC+L-GM | 40× | 0.9517 | 0.9524 | 0.9520 | 0.9714 | 0.8872 | 0.9293 | - | - | - | - |
| | 20× | 0.9702 | 0.9702 | 0.9702 | 0.9811 | 0.9277 | 0.9544 | - | - | - | - |
| | 40×+20× | 0.9723 | 0.9725 | 0.9724 | **0.9815** | **0.9296** | **0.9556** | **0.7111** | **0.9681** | **0.6121** | **0.9279** |

generated by DMC are closer to the ground truth. Regarding ROC AUC of pixel level, the heat map generated by DMC with L-GM loss is also the best. On the WSIs of the training set, DMC performs similarly to the single input classifier under 40×. But the score of DMC is 2.5% higher than the classifier under 40× on the WSIs of the test set. We think that the higher the AUC, the better the selection and prediction of sampling points.

## Evaluating adaptive sampling algorithms

In this section, we mainly evaluate the performance of the sampling algorithms by comparing the probability heatmaps using the same DMC. As before, the measures used for comparing the algorithms were the F1 score and AUC at the pixel level.

Table 3 shows the pixel detection performance comparison between HASHI and our sampling method on tumor samples. In the parameters of HASHI, the number of samples per iteration was fixedly set to 400, and the maximum iterations $T$ was set to 20, 30, and 40, respectively. In our sampling method, we evaluated both non-post-processing and post-processing (The slide filter was marked as 'SF' in Table 3). We report the average F1 score and AUC for these approaches with the Camelyon16 and HCH datasets. Here the experiment WSIs were divided into three groups: Camelyon16 Train, Camelyon16 Test, and HCH Test. The patches for training the DMC classifier were extracted from WSIs of training data of Camelyon16. In other words, a part of the sampled patches may exist in the training set. So, the classifier had higher classification performance for such WSIs. The other two datasets were never seen by the patch-based classifier.

Compared to the F1 score of the patch level, the score of the pixel level did have a significant decline. On the other hand, AUC at pixel level was still relatively high, and that of our method exceeded 0.95 on all datasets. Because the DMC classifier was trained on labeled patches and had not been trained using pixel-level labeled data on WSIs. Therefore, the probability heatmap is better in overall probability prediction, but the contours of the probability of 0.5 at the heat map were still a little bit different from the ground truth.

For HASHI, when the accuracy of the classifier is sufficient on Group Camelyon16 Train, both F1 and AUC will increase as the number of sampling points increases. On the other verification groups, F1 and AUC did not improve even if the number of sampling points was doubled. Because for the verification groups, the F1 score of the tumor patches was 0.9296, which was 0.04 lower than that of the training set in Table 2.

Compared with HASHI, our proposed adaptive sampling method has better results. The F1 and AUC of our method with post-processing are the highest of all tests. Our F1 score is at least 5.8% higher than its predecessors, and AUC is at least 3.2% higher than that. Regarding the slide filter, the post-processing has a 1.6% improvement on F1 and 1.9% on AUC. It is worth noting that our F1 and AUC without slide filter were not significantly different from

**Table 3. The pixel-level detection performance on different sampling algorithms with DMC classifier.**

| Methodology | Camelyon16 Train | | Camelyon16 Test | | HCH Test | | Number of sampling points |
|---|---|---|---|---|---|---|---|
| | F1 | AUC | F1 | AUC | F1 | AUC | |
| Our method | 0.7111±0.1839 | 0.9681±0.0782 | 0.6121±0.2631 | 0.9279±0.1028 | 0.6999±0.2041 | 0.9342±0.0485 | 7402±2028 |
| Our method2* | **0.7113**±0.1813 | **0.9782**±0.0524 | **0.6173**±0.2720 | **0.9527**±0.0819 | **0.7439**±0.1929 | **0.9577**±0.0342 | 7402±2028 |
| HASHI T = 20 | 0.5879±0.2633 | 0.9393±0.0978 | 0.5695±0.3090 | 0.8782±0.1815 | 0.6810±0.2269 | 0.9451±0.0424 | 8000 |
| HASHI T = 30 | 0.6129±0.2682 | 0.9660±0.0491 | 0.5425±0.3317 | 0.8451±0.2259 | 0.6844±0.2257 | 0.9415±0.0447 | 12000 |
| HASHI T = 40 | 0.6424±0.2379 | 0.9690±0.0425 | 0.5574±0.3122 | 0.8787±0.1864 | 0.6832±0.2311 | 0.9409±0.0454 | 16000 |

*Our method2 is the method using Slide Filter in postprocessing.

HASHI on Group HCH. This was not the case with Group Camelyon16 Test. This is because the area of the tumor regions in each slide of HCH is on average 8 to 9 times larger than Camelyon16, but the number of regions is generally relatively small. In other words, the detection target is relatively significant. Therefore, HASHI is more suitable for the detection of such WSIs, and our proposed method can detect more and smaller tumor regions.

### Two evaluation in Camelyon16

In this section, we briefly present two evaluation results in Camelyon16: Slide-based Evaluation and Lesion-based Evaluation.

**Slide-based evaluation.** This evaluation task is to distinguish between slides containing metastasis and normal slides and rank them by the area under ROC curve (AUC) [6]. For the slide-based classification task, the postprocessing method takes a prediction result for each WSI as input and produces a single probability of tumor for the entire WSI as output. Here, we extracted 5 statistical features from the positive part of the predictions $\mathcal{F}$, whose tumor probability is greater than 0.5. These features included the number of sample points that meet the probability requirements, the maximum tumor probability among them, and a normalized histogram with three bins based on these tumor probabilities. We computed these features over the predictions across all cases, and we trained and compared 4 classifiers to discriminate whether a WSI includes tumor regions. The merits of the algorithms will be assessed for discriminating between slides containing metastasis and normal slides. Receiver operating characteristic (ROC) analysis at the slide level will be performed, and the measure used for comparing the algorithms will be the area under the ROC curve (AUC) [37]. On the independent test cases, the Lagrangian-based $S^3VM$ [38, 39] model achieved an AUC of 0.9920, as shown in Fig 7. Our score is very close to the score (0.9935) of the top-ranked team on the leaderboard of the Camelyon16 ISBI challenge [37].

**Lesion-based evaluation.** The second evaluation task is to test the detection/ localization performance, which is summarized using free-response operating characteristic (FROC) curves [37]. For lesion-based detection, a pair probability and corresponding coordinate of each predicted cancer lesion within the WSI need to be given with few false positives. Our approach is similar to [40], which used a non-maxima suppression method. In contrast, we used the Isolation Forest algorithm [41] and the K-means (K = 2) clustering to find automatic segmentation thresholds for a tumor probability heatmap.

The FROC curve is defined as the plot of sensitivity versus the average number of false positives per image [37]. As shown in Fig 8, our method achieved a score of 0.7694 at 1 FP per WSI on the training cases and a score of 0.7373 at 1 FP on the test cases. Table 4 shows the comparison with other methods using Camelyon16. Our score has reached the level of human performance. However, there is still a large gap between the current best score (0.8533), which was achieved by Fast ScanNet [42]. Fast ScanNet used a fully convolutional network without an up-sampling path to generate a probability heatmap with a much smaller size than the input image, then performed a dense scan on ROIs and stitched the predictions into a complete heatmap. The label of the patch-based training sample of Fast ScanNet was pixel-level and our classifier used patch level, it is not difficult to understand that the former performed well in FROC of the pixel-level detection. From our heat map, our proposed method usually combines multiple smaller tumor regions into one larger region for reporting. In the FROC measurement, this caused many small tumor regions to be detected but not reported. Another phenomenon is that the F1 score is not high (<0.75) and the AUC is high (>0.95) in Table 3.

On the other hand, Fast ScanNet still needs to completely scan the entire area to be detected, but our adaptive sampling algorithm does not need to do this. At the same time, Fast

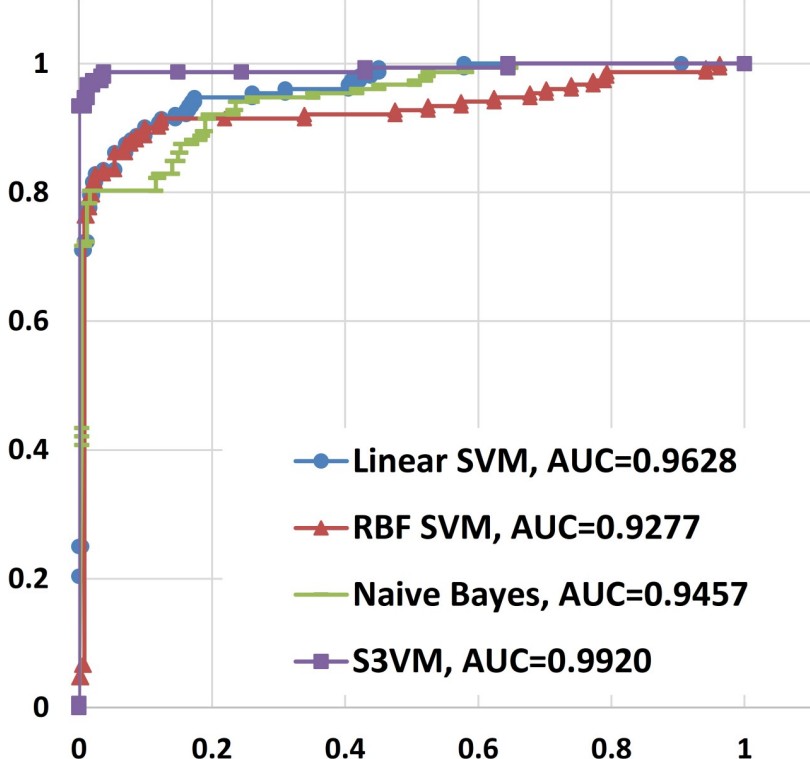

**Fig 7. Receiver Operating Characteristic (ROC) curve of slide-based classification.**

ScanNet used an FC net to generate heat maps, and a large number of large-size (2866×2866) feature maps were produced during the convolution process. Therefore, it put forward higher requirements for the memory capacity of GPU. And our method only needs to use 256×256 input images, as long as your computer can run PyTorch, you can complete the WSI detection task. When computing resources are limited, our proposed algorithm is a feasible and effective method.

Next, we discuss this issue in detail in the next section.

## Model runtime efficiency

Due to the use of a lightweight network, the computational complexity is relatively low, and approximately 286.2 pairs of double-magnification patches can be predicted per second. The performance test was performed on a PC with a 3.2 GHz Intel i7-8700 CPU with 16 GB of memory and an NVIDIA GeForce GTX 1080 8 GB.

The core of our proposed model is the DMC classifier, which is called thousands of times. However, it only contains 306,498 parameters. Compared with the patch-based classic network, the parameter size of VGG16 is 460 times that of our model, the size of GoogLeNet is 80 times, the size of ResNet-50 is 85 times, and the size of DenseNet-121 is 27 times. Therefore, the DMC classifier only takes one second to predict nearly 300 pairs of 256×256 patches from the saved small JPG files.

A heat map of a WSI under 1.25× contains an average of 15.1 million pixels. When a full dense scan of a WSI is performed at equal intervals without coverage under 20×, it is necessary to extract and predict approximately 59,000 patches of size 256×256. As shown in Table 3, when the number of samples per iteration (400) was fixed in HASHI, the number of extracted

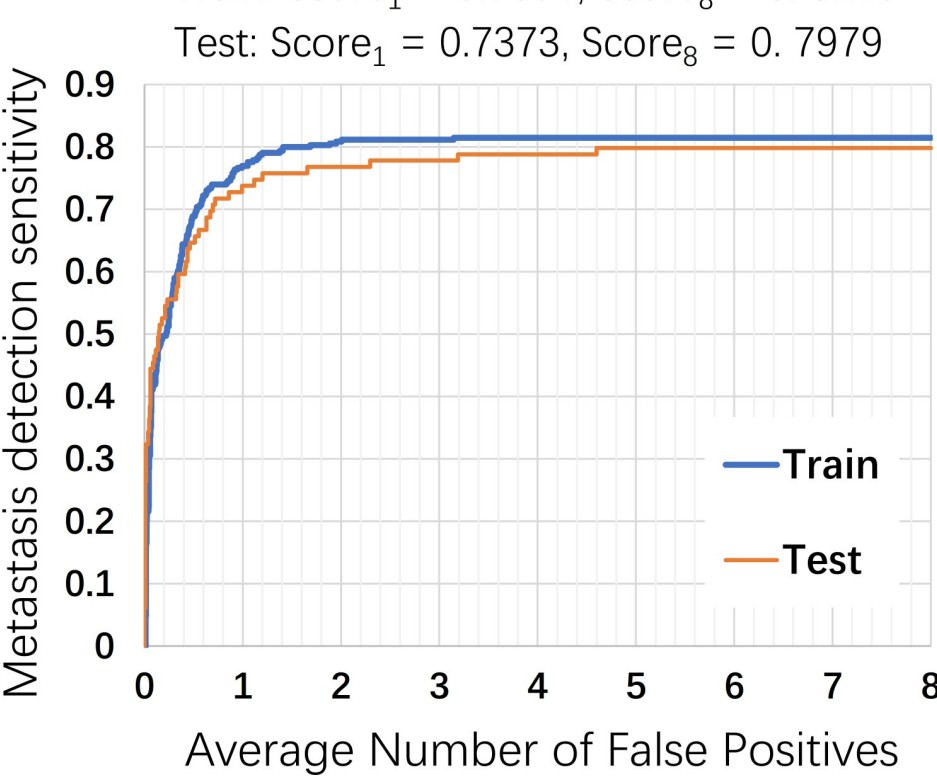

**Fig 8. FROC curve of the lesion-based detection.**

patches of per WSI is directly proportional to the maximum iterations $T$. When T was equal to 20, HASHI here needed to predict 8000 samples, which accounted for 13.6% of the number of the full dense scan. Our algorithm uses a cluster-based heuristic factor to select sampling points, instead of selecting the larger half ones of the gradient in HASHI. For scanning a slide, the number of patches extracted by our sampling algorithm is different for each slide.

**Table 4. Detection performance comparison with Camelyon16.**

| Team | AUC | FROC |
|---|---|---|
| Human performance | 0.9660 | 0.7325 |
| HMS and MIT | **0.9935** | 0.8074 |
| **Our method** | 0.9920 | 0.7373 |
| Fast ScanNet-16 | 0.9875 | **0.8533** |
| HMS, Gordon Center, MGH | 0.9763 | 0.7600 |
| CUHK | 0.9415 | 0.7030 |
| EXB Research | 0.9156 | 0.5111 |
| DeepCare, Inc. | 0.8833 | 0.2430 |
| Middle East Tech. Uni. | 0.8632 | 0.3822 |
| NLP LOGIX Co. | 0.8298 | 0.3859 |
| Smart Imaging Tech. Co. | 0.8207 | 0.3385 |
| Univ. of Toronto | 0.8149 | 0.3822 |
| Radboud Uni. | 0.7786 | 0.5748 |

Typically, about 7400 samples are extracted for each slide by our method, which accounted for 9.1 ~ 15.9% of the number of the full dense scan. The computational complexity using the local sampling density ($T_\rho = 1$) is equivalent to HASHI using the parameter $T = 20$. In Table 3, the detection results of our method are better. On the WSIs of the training set, the F1 score and AUC are 12.3% and 3.9% higher than HASHI. On the WSIs of the two test sets, the F1 score and AUC have improved by 5.5% and 4.3% on average compared to HASHI.

In time consumption, our method usually takes 2 to 3 minutes to complete a WSI with i7 CPU and single GTX 1080 8 GB. Our proposed method reduces the detection area by at least 85% in the adaptive sampling manner and saves the computing load of each sampling point with the lightweight network. Through the divide-and-conquer approach, the need for the memory capacity of GPU is drastically reduced, and at the same time, detection effects can meet the needs of preliminary screening in clinical diagnosis.

## Conclusion

We proposed a novel lightweight detection framework for automatic tumor detection in whole-slide histopathology images. Compared to classic CNN models, our DMC model with dual inputs and three outputs is easier to train, with higher computational efficiency with only 0.3 million parameters. Our improved adaptive sampling method uses a new heuristic factor as the convergence condition of iterations for improving the detection performance of multiple instances, which is only 1/8 of the workload of the uncovered regular sampling. In post-processing, the patch-level predictions are regulated based on the predictions of adjacent sampling points to improve the pixel level and lesion level accuracy. Our experiments revealed that our method also has reached the state of the art on the pixel level and lesion level detection of gigapixel pathology slides with limited computing resources. In clinical practice, the ability to use more computer resources for detecting whole-slide images will greatly promote the practical application of automatic diagnostic technology.

With the continuous popularization of breast cancer screening, more and more early-stage breast cancers containing carcinoma in situ have been discovered. On whole-slide images, how to accurately identify the presence and proportion of carcinoma in situ and invasive cancer is extremely important for selecting the appropriate treatment and the best benefit for the patient. In future work, we aim to study region detection of carcinoma in situ and invasive cancer. We could explore a new clustering model for encoding histology WSI to analyze the texture features on tissue structure in a larger field of view.

## Supporting information

**S1 Appendix.**
(DOCX)

## Author Contributions

**Data curation:** Junqiu Yue.

**Funding acquisition:** Junqiu Yue.

**Methodology:** Jun Ruan.

**Project administration:** Junqiu Yue.

**Software:** Jun Ruan, Zhikui Zhu, Chenchen Wu, Guanglu Ye, Jingfan Zhou.

**Validation:** Zhikui Zhu, Chenchen Wu, Guanglu Ye, Jingfan Zhou.

**Writing – original draft:** Jun Ruan.

**Writing – review & editing:** Jun Ruan, Junqiu Yue.

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
