## [Decision Letter · Decision Letter 0]

4 Mar 2021

PONE-D-21-02087

A fast and effective detection framework for Whole-Slide Histopathology Image analysis

PLOS ONE

Dear Dr. Yue,

Thank you for submitting your manuscript to PLOS ONE. After careful consideration, we feel that it has merit but does not fully meet PLOS ONE’s publication criteria as it currently stands. Therefore, we invite you to submit a revised version of the manuscript that addresses the points raised during the review process.

The manuscript had been reviewed by 2 reviewers. Reviewer 1 was of the view that manuscript partly describes a technically sound piece of scientific research and recommended major revision. Reviewer 2 was of the opinion that your manuscript describe technically piece of scientific research, however, he had made certain observations and also recommended major revision. 

After thorough consideration of comments of Reviewer 1 and Reviewer 2, my decision is "major revision". Please incorporate comments raised by both reviewers. 

**Additional AE Note Authors**: I have noted that one of the reviewers has asked for more context in the literature review, and suggested specific papers to be cited. While you may take on-board their suggested papers if you feel that they are relevant for your manuscript, or just take on-board the general suggestion for providing some more context in the literature review, there is no requirement from the journal to cite these papers

We look forward to receiving your revised manuscript.

Kind regards,

Gulistan Raja

Academic Editor

PLOS ONE

Journal Requirements:

Reviewers' comments:

Reviewer's Responses to Questions

**Comments to the Author**

1. Is the manuscript technically sound, and do the data support the conclusions?

Reviewer #1: Partly

Reviewer #2: Yes

2. Has the statistical analysis been performed appropriately and rigorously? 

Reviewer #1: Yes

Reviewer #2: Yes

3. Have the authors made all data underlying the findings in their manuscript fully available?

Reviewer #1: Yes

Reviewer #2: Yes

4. Is the manuscript presented in an intelligible fashion and written in standard English?

Reviewer #1: No

Reviewer #2: No

5. Review Comments to the Author

Reviewer #1: In this paper, a “A fast and effective detection framework for Whole-Slide Histopathology Image analysis” is introduced, where a Double Magnification Combination (DMC) classifier, which is a modified DenseNet-40 to make patch-level predictions with only 0.3 million parameters. The detailed comments are given below from nine respects:

1. Language: The language is OK for me to understand the authors idea. However, there are still many grammar problems, such as Line 29, 30, 31, 35, 41-42, 46, 81-82, 97, 102, 105-107. Except these problems, there are still many others to improve. In total, the language quality of this paper is very poor and need to improve quite a lot for a final publication.

2. Application value This work focus on the histopathology image analysis, which is meaningful to support the cancer clinick work in practical. This is good.

3. Scientific value and novelty: The scientific value of this work is somehow limitied by the novelty of this work. In my opinion, although it said that this work developed some new method, it is a combination of some exsiting famous methods in the histopathology image analysis domain. Please explain more detailes: (1) in Line 134, what is the “simple threshold-based segmentation method”? (2) in Line 155-156, “For the choice of patch-based classifier structure, we chose nine classic pre-trained ImageNet networks to test the patch-based 156 classifier under three magnifications”, what is the motivation to use these methods? (3) in Line 162-163, “Our modified network has only 0.3 million parameters with 3 dense blocks defined in DenseNet, and 163 each block consists of 12 convolution layers. The growth rate (‘k’ in [17]) is 16”, what did authors use Densenet40? Due to the normally used structure Densenet is Densenet41, lease give the structure of Densenet40. Why did the authors set growth rate k=16?

4.Experimental data: based on my research experiences in the histopathology image analysis domain, the quality and data size in this paper is OK.

5. Experimental result: in this paper, a large number of contrast experiments were carried out to show the effectiveness of the new method, and good experimentals results were obtained.

6. Figure quality: Figure 1-6 have low quality to improve.

7. Table quality: the layouts of some tables are uncomfortable to read. Please update.

8.Euqation quality: I can understand most of the equations in this paper, and did not find obvious problems. However, in Line 181-199, these two loss functions are well-known, so the authors do not need to introduce them so detailed.

9. Reference quality: most references are ok, but some are too old and some new papers should be read and added.

[1] Gastric histopathology image segmentation using a hierarchical conditional random field （2020）

[2] RMDL: Recalibrated multi-instance deep learning for whole slide gastric image classification （2019）

[3] Superpixel-Based Conditional Random Fields (SuperCRF): Incorporating Global and Local Context for Enhanced Deep Learning in Melanoma Histopathology （2019）

[4] A review for cervical histopathology image analysis using machine vision approaches (2020)

[5] A Cervical Histopathology Image Clustering Approach Using Graph Based Features (2020)

[6] A Comprehensive Review of Markov Random Field and Conditional Random Field Approaches in Pathology Image Analysis (2021)

[7] Computerized Spermatogenesis Staging (CSS) of Mouse Testis Sections via Quantitative Histomorphological Analysis (2020)

Reviewer #2: Dear authors,

the research work is very interesting. The appendix is very useful. Globally, we miss the big picture in favor of a lot of technical details that should be better organized for a general audience. The methodology is basic but you bring good performances for time computing and recognition scores as well (on par with human discrepancy)

My detailed remarks /questions follow with citations of your text :

- 80 000 x 80 000 at 40x : 2G.

I am surprised with the size at 40x I would say 20 G ?

“Because the patches at the edge of annotations are usually transitional regions from tumor tissue to normal tissue, most of the “hard examples” are concentrated in these positions. The same is true for the labeling of patch pairs under double magnifications”

Are you sure ? For me, the border of the tumoral areas are quite visually marked even I agree there is a transition that makes it more interesting to be precise from a computational point of view.

Here, the sampling interval in the normal regions is larger than that in the tumor region.

Here stands for where ? What do you mean ?

because the prediction of a patch with the same size under 40× is more accurate for the center of the patch. ??

Rephrase it or explain it more clearly

The structure of the patch-based classifier was derived from DenseNet-40, so we called it the Double Magnification Combination (DMC) patched-based classifier.

So ?? I do not see the logical implication ?

The growth rate (‘k’ in [17]) is 16.

is set to 16 and explain what it is

Here, y refers to the ground truth under double magnification, y 20 and y 40 and so forth.

The prediction is the same in classif, isn’t it ?

Here, α is 1.0, and w is the weight of the L-GM loss, which is 0.001 by

explain alpha et w why 0,5 and w if w is set to a fixed value. It complexity the method for no added value. Or you make w varying ?

The predications ?

:-) predictions of course

Ca and Cr a and r stands for what

Globally as there is a lot of parameters make it clear the choice of subscript

Here, T f represents the lower limit of the tumor feature with a superpixel, generally set to -1, which indicates that a superpixel within a tumor probability of more than 27% should be considered

Make it clearer please??

Fig 4 is not very clear ?

the 12 points for instance

Here, we did not use fully-conv (FC) net to directly generate a heatmap under 1.25×, because this requ ired higher hardware requirements (GPU memory capacity).

But you use a GPU in the results ? (see “The performance test was performed on a PC with a 3.2 GHz Intel i7-8700 CPU with 16 GB of memory and an NVIDIA GeForce GTX 1080 8 GB”.)

When the features under both magnifications are used at the same time in DMC, the performance was improved by nearly 2~3%. Because a pair of patches overlap at the center point and the field of view is different, so the spatial attention mechanism was introduced. For the performance of patch-level detection, our experience is the use of L-GM loss in training has no significant effect on single-input or dual-input classifiers.

Results interesting : x20 seems to be enough

Is it very useful to add ‘0x then. ?

Although our training and testing sets contain millions of patches extracted from WSIs, the patches extracted during the adaptive sampling process were still more complex.

I do not understand the logical articulation.

Table 3 interesting

HCH very different from Camelyon : can you elaborate on it a bit more as you started to do in the following lines.

This is because the area of the tumor regions in each slide of HCH is on average 8 to 9 times larger than Camelyon16, but the number of regions is generally relatively small. In other words, the detection target is relatively significant. Therefore, HASHI is more suitable for the detection of such WSIs, and our proposed method can detect more and smaller tumor regions.

For instance, show HCH Images and the difference with Camelyons

Therefore, it put forward higher requirements for the memory capacity of GPU. And our method only needs to use 256×256 input image, as long as your computer can run PyTorch, you can complete the WSI detection task. When computing resources are limited, our proposed algorithm is a feasible and effective method.

The core of our proposed model is DMC classifier, which is called thousands of times.

Therefore, the DMC classifier only takes one second to predict nearly 300 pairs of 256×256 patches from the saved small JPG files.

This part is useful for the reproducibility. Many thanks.

Globally improve the quality of rendering of the Figures (resolution I guess ?)

6. PLOS authors have the option to publish the peer review history of their article (what does this mean?). If published, this will include your full peer review and any attached files.

Reviewer #1: No

Reviewer #2: No

---

## [Author Response · Author response to Decision Letter 0]

10 Apr 2021

Reviewer#1, Concern # 1: in Line 134, what is the “simple threshold-based segmentation method”?

Author response: For excluding the obvious background region, we apply three fixed-level thresholds in HSV to the thumbnail image of WSIs by experience. In the S channel, its value is required to be greater than 0.1. The value of the V channel is required to be between 0.2 and 0.8. These have been introduced in Line 135-136 of the original paper.

Reviewer#1, Concern # 2: in Line 155-156, “For the choice of patch-based classifier structure, we chose nine classic pre-trained ImageNet networks to test the patch-based classifier under three magnifications”, what is the motivation to use these methods?

Author response: We tried to find a suitable classifier structure using transfer learning. The features of patches were extracted by nine pre-trained models. And we replaced the original top layer with a new one to connect each feature extraction part, which consists of a Global Average Pool (GAP) and two fully-connected layers. We used the prepared patches on three different magnifications to fine-tuning the top layer of each transfer model and tested the accuracy of these 18 models. All testing results of transfer-learning are shown in Supplementary Table 1. According to the results of transfer-learning, the DenseNet family has the best performance of feature extraction for pathological image blocks. Therefore, we construct our network based on DenseNet.

Supplementary Table 1 Accuracy of patch-based classification on different magnification using different Transfer models

Model name Parameters

(M) Image size Binary Accuracy

 10× 20× 40×

Inception_v3 23.9 224x224x3 0.9231 0.9573 0.8835

DenseNet121 8.1 299x299x3 0.9329 0.9640 0.8946

DenseNet169 14.3 299x299x3 0.9273 0.9656 0.8978

DenseNet201 20.2 299x299x3 0.9329 0.9651 0.8946

ResNet50 25.6 299x299x3 0.8604 0.9300 0.8146

Inception_ResNet_v2 55.9 224x224x3 0.9218 0.9599 0.8904

VGG16 138.4 299x299x3 0.9186 0.9537 0.8757

MobileNet_v2 3.5 299x299x3 0.9063 0.9506 0.8752

NASNet(mobile) 5.3 299x299x3 0.9100 0.9595 0.8849

We revised the manuscript accordingly, and the above table has been added to the supplementary materials.

Reviewer#1, Concern # 3: in Line 162-163, “Our modified network has only 0.3 million parameters with 3 dense blocks defined in DenseNet, and 163 each block consists of 12 convolution layers. The growth rate (‘k’ in [17]) is 16”, what did authors use Densenet40 ? Due to the normally used structure DenseNet is Densenet41, lease give the structure of Densenet40. Why did the authors set growth rate k=16?

Author response: Our network consists of three dense blocks defined in DenseNet. Each block consists of 6 dense layers that each contains two convolution layers. Between two adjacent dense blocks, there is a translation layer that consists of one convolution layer. And only two transport layers are used here. The network also contains a convolutional layer at the input and a fully connected layer at the top. Therefore, we refer to this network architecture as DenseNet40 (=3x6x2+2x1+1+1=40).

Here we set the growth rate k to 16 to reduce the parameters of the model by using very narrow layers, at the same time, keeping up with the performance of patch classification.

In the corresponding section of the revised manuscript, we provide a detailed description.

Reviewer#1, Concern # 4: in Line 181-199 , these two loss functions are well-known, so the authors do not need to introduce them so detailed.

Author response: We very much appreciate the comments. We have trimmed this section.

Reviewer#1, Concern # 5: most references are ok, but some are too old and some new papers should be read and added.

[1] Gastric histopathology image segmentation using a hierarchical conditional random field （2020）

[2] RMDL: Recalibrated multi-instance deep learning for whole slide gastric image classification （2019）

[3] Superpixel-Based Conditional Random Fields (SuperCRF): Incorporating Global and Local Context for Enhanced Deep Learning in Melanoma Histopathology （2019）

[4] A review for cervical histopathology image analysis using machine vision approaches (2020)

[5] A Cervical Histopathology Image Clustering Approach Using Graph Based Features (2020)

[6] A Comprehensive Review of Markov Random Field and Conditional Random Field Approaches in Pathology Image Analysis (2021)

[7] Computerized Spermatogenesis Staging (CSS) of Mouse Testis Sections via Quantitative Histomorphological Analysis (2020)

Author response: Thank you for the suggestion about these references. We updated the introduction section by explaining the recent studies related to WSI analysis, as suggested by the reviewer. Since the manuscript focuses on the classification and localization of tumor regions at the pixel level on a WSI, we do not cite all the above literature. We hope that the reviewer will be satisfied.

Reviewer#1, Concern # 6: Figure 1-6 have low quality to improve. And the layouts of some tables are uncomfortable to read. Please update.

Author response: Thank you very much for the feedback about the quality of the figures and tables. We will try our best to improve.

Reviewer#1, Concern # 7: There are still many grammar problems, such as Line 29, 30, 31, 35, 41-42, 46, 81-82, 97, 102, 105-107. Except these problems, there are still many others to improve.

Author response: We very much appreciate the comments. We tried to revise the paper. We hope the English grammar would be better now. We also hope that we answer the reviewer questions properly.

Reviewer#2, Concern # 1: 80 000 x 80 000 at 40x : 2G. I am surprised with the size at 40x I would say 20 G ?

Author response: Whole-slide images are stored in a multi-resolution pyramid structure. Image files contain multiple down-sampled versions of the original image. Each image in the pyramid is stored as a series of tiles, to facilitate rapid retrieval of subregions of the image. In other words, the WSIs have been highly compressed. For example, the compressed storage space of the No. 76 tumor sample is 2.24GB with 114688 x 100352 pixels at 40x, but the uncompressed size is 32.2GB. 

Reviewer#2, Concern # 3: “Because the patches at the edge of annotations are usually transitional regions from tumor tissue to normal tissue, most of the “hard examples” are concentrated in these positions. The same is true for the labeling of patch pairs under double magnifications”

Are you sure ? For me, the border of the tumoral areas are quite visually marked even I agree there is a transition that makes it more interesting to be precise from a computational point of view.

Author response: We apologize for the inaccurate description and confusion here. In the latest version, the labeling method has been adjusted. The labeling of a patch is determined by the proportion of the tumor area within the patch. When the proportion of tumor area is less than 50%, this patch is normal (Label 0); otherwise, it is tumor (Label 1). For the labeling under the combination of double magnifications, we adopted the "or" logic here. 

Compared with the method of judging whether the center point of a patch is in the ground truth annotation, these updated labels make the patch-based classifier more robust. The accuracy of the predicted segmentation boundary is attempted to be ensured by our adaptive algorithm, which can perform further sampling in suspicious regions or boundaries.

The experimental results in the manuscript are obtained using the above-mentioned lastest annotation method. In the revised version, we made corresponding corrections.

Reviewer#2, Concern # 4: Here, the sampling interval in the normal regions is larger than that in the tumor region.

Here stands for where ? What do you mean ?

Author response: Sorry for not expressing clearly here. We balanced the number of positive and negative samples in a WSI by dynamically changing the sampling interval of different regions. If the area of normal regions in a WSI is much larger than the tumor area, we will increase the sampling interval in the normal regions. In this way, the number of negative samples in a WSI does not exceed 5~6 times the number of positive samples. Then, we constructed balanced training sample sets (1:1) through random sampling for improving the performance of the patch-based classifier. We have corrected this part in the manuscript.

Reviewer#2, Concern # 5: because the prediction of a patch with the same size under 40× is more accurate for the center of the patch. ?? 

Rephrase it or explain it more clearly

Author response: 

We use patches of the same image size under different magnifications, and the centers of the two patches under different magnifications are coincident. According to the experimental results of transfer learning in Supplementary Table 1, we observed the following phenomena.

The patches under 20× have the best distinguishable characteristics which can be extracted by CNN, as a result of the balance of texture details and texture range in view. Although the 10× patches have a larger field of view, they are downsampled to the same size resulting in the loss of texture and degradation of classification performance. Compared to 20×, the classifiers under 10× are a little worse. Under 40×, the field of view in a patch becomes very small in a patch. When the patches are extracted from the transitional zone from tumor to normal near the edge of annotations, these patches under 40× are no significant and typical texture features, and even look the same as the patches in normal regions. So, it is difficult to train a better classifier under this magnification individually. On the other hand, the patch-based classifier calculates a tumor feature based on the entire 256 × 256 image, and the calculated patch-level prediction is stored in a tumor feature map based on the central coordinate of the patch.

Thus, with the same image size, the prediction under higher magnification can more accurately represent the tumor feature (probability) at the sampling point (the center of a patch). From the perspective of spatial location, we argue that the prediction of a patch with the same size under 40× is more accurately express the tumor feature at the center of the patch, and facilitate the generation of more detailed segmentation boundaries. Moreover, in the pixel segmentation experiment, the accuracy under the 20× and 40× magnifications alone is better than that under the 10× alone. In the end, our patch-based classifier chose the combined input under 20× and 40×.

Reviewer#2, Concern # 6: The structure of the patch-based classifier was derived from DenseNet-40, so we called it the Double Magnification Combination (DMC) patched-based classifier.

So ?? I do not see the logical implication ?

Author response: We apologize for the illogical expression here; this “so” is redundant. 

Reviewer#2, Concern # 7: The growth rate (‘k’ in [17]) is 16. is set to 16 and explain what it is

Author response: The growth rate is a hyper-parameter of the DenseNet model. In a dense block, from input to output, the number of feature-maps of the convolutional layers continues to increase, and the increment is the growth rate mentioned here. 

We set the growth rate to 16 to reduce the parameters of the model by using very narrow layers, at the same time, keeping up with the performance of our patch classification.

Reviewer#2, Concern # 8: Here, y refers to the ground truth under double magnification, y 20 and y 40 and so forth. The prediction is the same in classif, isn’t it ?

Author response: As mentioned earlier, the labeling of a patch is determined by the proportion of the tumor area within the patch. When the proportion of tumor area is less than 50%, this patch is normal (Label 0); otherwise, it is tumor (Label 1). For the labeling under the double magnification combination, we adopted the "or" logic here.

Thus, inside the normal and tumor regions, the labels y20 and y40 of a pair of patches overlap at the center point are the same. The y20 and y40 of a pair of patches extracted from the edges of manual annotations may be different. And the label y takes their logical union.

Reviewer#2, Concern # 9: Here, α is 1.0, and w is the weight of the L-GM loss, which is 0.001 by explain alpha et w why 0,5 and w if w is set to a fixed value. It complexity the method for no added value. Or you make w varying ?

Author response: α is a hyper-parameter of L-GM loss. In the reference, its default value is 1. Here, we followed this setting. 

w is the weight of the regularization term of the loss function 〖loss﷩2〗, and It is an empirical value.

The first factor of 0.5 in 〖loss〗_2 indicates that the two cross-entropy losses at both magnifications have the same weight, that is, the classification error is reflected in their average under both magnifications.

Reviewer#2, Concern # 10: Ca and Cr a and r stands for what 

Globally as there is a lot of parameters make it clear the choice of subscript

Author response: We very much appreciate the comments. In the revised manuscript, we tried our best to clarify the meaning of these symbols.

In the pseudo-code of Algorithm 1, C_R refers to the set of center coordinates of patches, which are obtained by regular sampling based on superpixel segmentation. The subscript "R" here is the initials of “Regular”. 〖C﷩A〗 refers to the set of center coordinates of patches, which are obtained by random sampling (Quasi-Monte Carlo) process. The subscript "A" here is the second letter of “random”, since "R" has already been used before. 

Reviewer#2, Concern # 11: Here, T f represents the lower limit of the tumor feature with a superpixel, generally set to -1, which indicates that a superpixel within a tumor probability of more than 27% should be considered 

Make it clearer please??

Author response: Because the output of our binary patch-based classifier is the tumor feature of a patch, if we use the Sigmoid function to regress a feature into a tumor probability, the feature value -1 corresponds to the tumor probability of 27%. Here the threshold T_f^sp represents the lower limit of the tumor feature in a superpixel, generally set to -1. When there is a feature larger than T_f^sp in a superpixel, it means that there is a point with tumor probability greater than 27% inside. In the next iteration, such superpixels will be further explored. 

We have revised this in the manuscript.

Reviewer#2, Concern # 12: Fig 4 is not very clear ?

the 12 points for instance

Author response: We very much appreciate the comments. We will try our best to improve.

Reviewer#2, Concern # 13: Here, we did not use fully-conv (FC) net to directly generate a heatmap under 1.25×, because this required higher hardware requirements (GPU memory capacity).

But you use a GPU in the results ? (see “The performance test was performed on a PC with a 3.2 GHz Intel i7-8700 CPU with 16 GB of memory and an NVIDIA GeForce GTX 1080 8 GB”.)

Author response: Due to limited resources, we only put the training and prediction of the DMC patch-based classifier to run on GPU. While in our adaptive sampling, the generation of sampling coordinates, the operations of feature maps M_feat and gradient maps M_grad are all done on the CPU. 

Reviewer#2, Concern # 14: When the features under both magnifications are used at the same time in DMC, the performance was improved by nearly 2~3%. Because a pair of patches overlap at the center point and the field of view is different, so the spatial attention mechanism was introduced. For the performance of patch-level detection, our experience is the use of L-GM loss in training has no significant effect on single-input or dual-input classifiers.

Results interesting : x20 seems to be enough 

Is it very useful to add ‘0x then. ?

Reviewer#2, Concern # 15: Although our training and testing sets contain millions of patches extracted from WSIs, the patches extracted during the adaptive sampling process were still more complex.

I do not understand the logical articulation.

Author response: These two issues are related and we give clarification here together.

Pathologists usually check images by changing their magnification and scope in the WSI. Ways to use images under a wider range of magnifications are worth studying. Inspired by this, we investigated the patch-based classifier with multiple magnifications. Regarding the F1 scores in Table 2, there is not much difference in accuracy between single-input or dual-input patch-based classifiers, but there is a significant difference in the results of pixel-level segmentation. In the pixel-level segmentation task, the F1 scores of each patch-based classifier are much lower than the scores of the classifier during training and testing. The loss of the pixel-level segmentation task is not used to optimize the performance of patch-based classifiers; it represents the generalization performance of the classifier. This is because, although we extracted millions of patches from WSI for training these patch-based classifiers, the input images during our adaptive random sampling are almost impossible to be the same as those in the training set. In other words, the patches extracted during the adaptive sampling are more varied. Moreover, the prediction error at any sampling point has an impact on the accuracy of the segmentation boundary near it. The superposition effect brought by the sampling mechanism makes it possible to obtain correct results only when the robustness of the patch-based classifier is sufficient. Regarding the F1 score of pixel level, the performance of our adaptive sampling algorithm on DMC is nearly 20% higher than that of the classifiers with single magnification. This shows the advantages of the dual input structure.

Reviewer#2, Concern # 16: Table 3 interesting

HCH very different from Camelyon : can you elaborate on it a bit more as you started to do in the following lines.

This is because the area of the tumor regions in each slide of HCH is on average 8 to 9 times larger than Camelyon16, but the number of regions is generally relatively small. In other words, the detection target is relatively significant. Therefore, HASHI is more suitable for the detection of such WSIs, and our proposed method can detect more and smaller tumor regions.

For instance, show HCH Images and the difference with Camelyons

Author response: We very much appreciate the comments. In the revised manuscript, we have made additions in Fig. 6.

---

## [Decision Letter · Decision Letter 1]

28 Apr 2021

A fast and effective detection framework for Whole-Slide Histopathology Image analysis

PONE-D-21-02087R1

Dear Dr. Yue,

We’re pleased to inform you that your manuscript has been judged scientifically suitable for publication and will be formally accepted for publication once it meets all outstanding technical requirements.

Kind regards,

Gulistan Raja

Academic Editor

PLOS ONE

Additional Editor Comments (optional):

Reviewers' comments:

Reviewer's Responses to Questions

**Comments to the Author**

1. If the authors have adequately addressed your comments raised in a previous round of review and you feel that this manuscript is now acceptable for publication, you may indicate that here to bypass the “Comments to the Author” section, enter your conflict of interest statement in the “Confidential to Editor” section, and submit your "Accept" recommendation.

Reviewer #1: All comments have been addressed

Reviewer #2: All comments have been addressed

2. Is the manuscript technically sound, and do the data support the conclusions?

Reviewer #1: Yes

Reviewer #2: Yes

3. Has the statistical analysis been performed appropriately and rigorously? 

Reviewer #1: Yes

Reviewer #2: Yes

4. Have the authors made all data underlying the findings in their manuscript fully available?

Reviewer #1: Yes

Reviewer #2: Yes

5. Is the manuscript presented in an intelligible fashion and written in standard English?

Reviewer #1: Yes

Reviewer #2: Yes

6. Review Comments to the Author

Reviewer #1: This paper was revised well.

However, the language quality is still not enough.

Please ask some native English language speakers to help to improve.

Reviewer #2: I understand then all your choices concerning the strategy.

So you work on compressed image. I think it is important to express it And which compression method or format if not tiff.

I would put Re and Ra then for the subscript in Algo 1 (for Regular and Random)

I would make a precision about the use of GPU only for training.

My comment about the x10, x20, x40 concerning the field of view and the image size : you should have worked at the real size in micrometers. Anyway the results is interesting as the CNN are usually built for fixed size images (in pixels).

The multiresolution issue is still hardly tackled with these techniques, true.

7. PLOS authors have the option to publish the peer review history of their article (what does this mean?). If published, this will include your full peer review and any attached files.

Reviewer #1: No

Reviewer #2: **Yes: **Nicolas Loménie

---

## [Editor Report · Acceptance letter]

3 May 2021

PONE-D-21-02087R1 

A fast and effective detection framework for Whole-Slide Histopathology Image analysis 

Dear Dr. Yue:

I'm pleased to inform you that your manuscript has been deemed suitable for publication in PLOS ONE. Congratulations! Your manuscript is now with our production department. 

Kind regards, 

on behalf of

Dr. Gulistan Raja 

Academic Editor

PLOS ONE